# Cloud-Based IoT Applications and Their Roles in Smart Cities

Tanweer Alam

Faculty of Computer and Information Systems, Islamic University of Madinah, Madinah 42351, Saudi Arabia; tanweer03@iu.edu.sa

**Abstract:** A smart city is an urbanization region that collects data using several digital and physical devices. The information collected from such devices is used efficiently to manage revenues, resources, and assets, etc., while the information obtained from such devices is utilized to boost performance throughout the city. Cloud-based Internet of Things (IoT) applications could help smart cities that contain information gathered from citizens, devices, homes, and other things. This information is processed and analyzed to monitor and manage transportation networks, electric utilities, resources management, water supply systems, waste management, crime detection, security mechanisms, proficiency, digital library, healthcare facilities, and other opportunities. A cloud service provider offers public cloud services that can update the IoT environment, enabling third-party activities to embed IoT data within electronic devices executing on the IoT. In this paper, the author explored cloud-based IoT applications and their roles in smart cities.

**Keywords:** smart cities; IoT; cloud computing; information processing; cloud-based IoT applications



## 1. Introduction

Cloud computing is the next phase in the advancement of internet-based computing, and it allows information technology capabilities to be used as a service. As smart devices move outside of the cloud infrastructure environment, the IoT can increase efficiency, performance, and throughput. Smart cities are residential regions that make systematic efforts to see for themselves the new location of records and communication technologies, achieve environmental sustainability, urban system authority, improved health, knowledge development, and network-driven advancement [1–3]. Cloud computing is the next phase in the growth of internet-based computing, allowing for the delivery of information and communication technology (ICT) resources through a network. In cloud infrastructure, the IoT can benefit from increased efficiency, performance, and payload. The presentation of cloud computing has supported the manner of development and dissemination, and industrial electronic business packaging. As a result, IoT and cloud are now very close to future internet technologies that are compatible with IoT systems.

The IoT is primarily concerned with challenges that arise in a dynamic and shared environment. IoT is a broad category that comprises of various adaptable and unusual devices with limited storage, power supplies, and performance capabilities. These constraints establish a barrier and impedance to the development of IoT systems, and include complex issues such as compatibility, efficiency, full functionality, and availability. One of the most promising methods that may be combined with IoT to overcome such limitations is cloud computing. The cloud provides shared resources (network, storage, computers, and software) distinguished by ubiquity, low cost, and aesthetic characteristics. This paper describes the existing communication, processing, and storage applications on a cloud-based IoT platform for smart cities. This platform may use cloud resources and services to gather, transfer, analyze, process, and store data. It may also use cloud resources and services to collect, transmit, search, analyze, and store data generated by complex scenarios. Figure 1 shows the cloud-based IoT platform to develop applications.

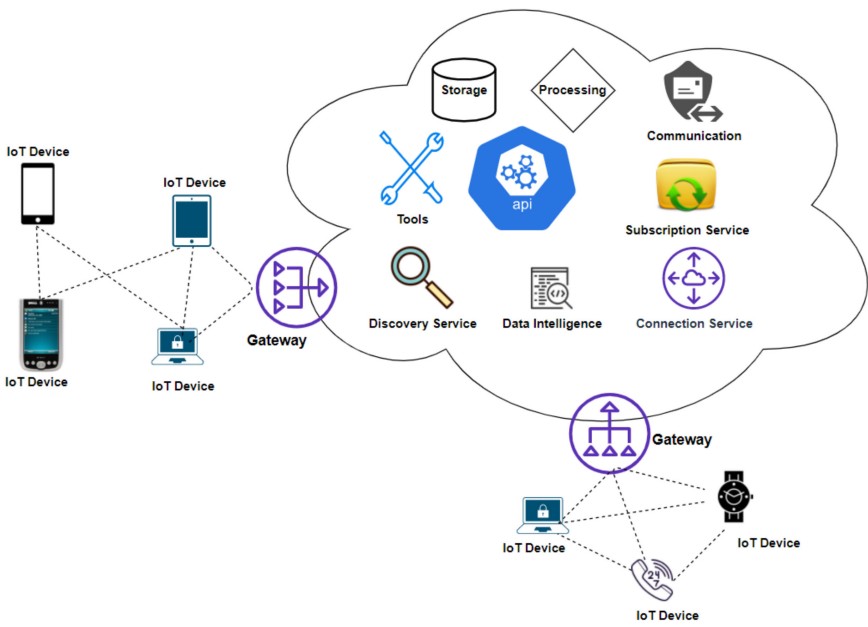

**Figure 1.** Cloud-based IoT System.

From industrial systems to emergency deliveries, public transportation, public safety, city lighting, and other metropolitan applications, the IoT has made its way into every commercial and public sector initiative. Cities are becoming connected as the IoT advances, allowing them to improve infrastructure installation efficiency and the reliability and responsiveness of emergency services. In the coming years, researchers are excited to explore new ideas for smart cities employing IoT solutions. Table 1 shows the related articles published between 2012–2021.

**Table 1.** Related articles.

| Authors | Ref. | Year | Title |
|---------|------|------|-------|
| Khan et al. | [1] | 2014 | Towards cloud-based smart cities, data security, and privacy management |
| Khan et al. | [2] | 2012 | A cloud-based architecture for citizen services in smart cities |
| Suciu | [3] | 2013 | Smart cities built on resilient cloud computing and secure IoT |
| Roy and Sarddar | [4] | 2016 | The Role of Cloud of Things in Smart Cities |
| Silva et al. | [5] | 2018 | Towards sustainable smart cities: A review of trends, architectures, components, and open challenges in smart cities |
| Chai et al. | [6] | 2021 | Role of BIC (Big Data, IoT, and Cloud) for Smart Cities |
| Rubí et al. | [7] | 2021 | An IoT-based platform for environment data sharing in smart cities |
| Kaur et al. | [8] | 2016 | Building smart cities applications using IoT and cloud-based architectures |
| Saleem et al. | [9] | 2020 | Building smart cities applications based on IoT technologies: A review |
| Dlodlo et al. | [10] | 2016 | Internet of things technologies in smart cities |
| Hyman et al. | [11] | 2019 | Secure controls for smart cities; applications in intelligent transportation systems and smart buildings |
| Curry et al. | [12] | 2016 | Smart cities–enabling services and applications |
| González-Zamar et al. | [13] | 2020 | IoT technology applications-based smart cities: Research analysis |
| Saravanan et al. | [14] | 2019 | Smart cities & IoT: evolution of applications, architectures & technologies, present scenarios & future dream |

**Table 1.** *Cont.*

| Authors | Ref. | Year | Title |
|---|---|---|---|
| Shamsir et al. | [15] | 2017 | Applications of sensing technology for smart cities |
| Saha et al. | [16] | 2017 | IoT solutions for smart cities |
| Song et al. | [17] | 2017 | Smart cities: foundations, principles, and applications |
| Sookhak et al. | [18] | 2018 | Security and privacy of smart cities: a survey, research issues, and challenges |
| Park et al. | [19] | 2018 | The role of IoT in smart cities: Technology roadmap-oriented approaches |
| Mehmood et al. | [20] | 2017 | Internet-of-things-based smart cities: Recent advances and challenges |
| Visvizi et al. | [21] | 2020 | Sustainable smart cities and smart villages research: Rethinking security, safety, well-being, and happiness |
| Talari et al. | [22] | 2017 | A review of smart cities based on the IoT concept |
| Delsing et al. | [23] | 2021 | Smart City Solution Engineering |
| Lanza et al. | [24] | 2016 | Smart city services over a future Internet platform based on IoT and cloud: The smart parking case |
| Syed et al. | [25] | 2021 | IoT in Smart Cities: A Survey of Technologies, Practices, and Challenges |
| Almalki et al. | [26] | 2021 | Green IoT for Eco-Friendly and Sustainable Smart Cities: Future Directions and Opportunities |

### 1.1. Cloud Is the Key for Internet-Based Computing

Cloud computing is the next phase in the growth of Internet-based computing, and it allows ICT services to be delivered as a carrier. Computer capabilities, infrastructure (e.g., servers and storage), systems, business processes, and other critical resources can all be connected via cloud computing [4]. The growth of cloud computing makes it easier to develop flexible business models, such as allowing organizations to use resources when their business grows. Unlike organizations that provide traditional web-based services (e.g., web hosting), cloud computing allows for immediate access to cloud delivery without a lengthy provisioning process. Each provisioning and withdrawal of resources in cloud computing can be repeated indefinitely. APIs (application programming interfaces) allow users to access cloud services and allow applications and resource records to communicate in the cloud. Invoicing and assessing providers are used in payment methods, providing the support required to use the rating assistance and to make payments in advance.

Monitoring and evaluating the performance: cloud computing infrastructure, in addition to the integrated system of physical computing and its methods, provides a carrier management environment to monitor and evaluate performance.

Security: cloud computing architecture offers secure operations that aim to protect sensitive information. The following are the two main business drivers behind the adoption of cloud computing and related services: (1) business enterprise. Cloud computing provides flexible, timely, and required access to computer resources as needed to fulfill business objectives; and (2) cost reduction. Cloud computing promises a cost reduction by converting capital expenditures (CapEx) into operating costs. This is because cloud computing allows for more flexible scheduling and resource allocation, and a preference for pre-existing management.

### 1.2. Six-Phases Startup Model for Building a Smart City

The following is a six-phase startup model for the formation of a smart city.

Phase 1: a fully smart city platform

The smart city implementation should start with a simple design. It will serve as the foundation for future development and allow you to add new services without losing overall performance [5]. The fundamental solution for smart cities consists of four requirements.

- A Network of Intelligent Objects

In smart city, IoT uses smart devices designed for sensors and actuators. The purpose of the sensory devices are to collect data and send it to a robust cloud control platform. Actuators allow devices to work, for example, by adjusting lights, reducing the flow of water in a pipe with leaks, and so on.

- Gateways for Secure Transmission

Every IoT device combines two elements: the hardware part of the IoT device and the applications part. Data cannot transmit from one object to another without applications. There should be gateways to control the data. These gates facilitate data collection and compression by filtering data before sending it to the cloud. The cloud gate ensures secure information transfer between local gates, and the cloud is part of the city's smart solution.

- Pool Facts

The main reason for the information pool is to keep records. Collections of data support evidence in smart cities. When data are requested, they are extracted from the pool and transferred to the destination.

- Large Record Warehouse

An extensive database is called a large records warehouse. Unlike statistical pools, it contains highly organized data. Once the actual data is determined, it is extracted, converted, and loaded into a large data warehouse.

Phase 2: data tracking and analytics

With data tracking and analytics, the process involves the collecting, recognizing, and classifying of data objects across the network system so that it may be used in data analysis. The technologies that corporations use to manage data, and the regulatory principles that they employ to safeguard consumer privacy and security, are all part of data tracking. For example, when reading data from soil moisture sensors planted in a park, cities can set digital valve rules to open or close at the identified humidity level. The information collected by the sensors can be seen in the dashboard of one platform, allowing customers to see all parts of the park.

Phase 3: analyze data

The amount of data produced by a community, transport networks, and digitization is incredible and continues to increase rapidly. With emerging IoT technologies, devices and cloud services significantly speed up the process of production. Analyzing, modelling, and extracting information from this data is a major contributor to the understanding of city contexts and can be used to improve the effectiveness of urban movement. Machine learning (ML) algorithms analyze ancient sensory data stored within an extensive database to determine progress and create predictable models. Models used by control packages send commands to IoT device actuators. Unlike the normal traffic modes designed to display a selected sign for a time period, smart visitors can adjust the entry times in traffic conditions. ML algorithms were developed on antiquated sensory systems to identify place visitor styles, control signal time, support average car speed acceleration, and avoid congestion.

Phase 4: smart control

Control systems ensure high automation of smart city devices by sending commands to their actuators. They "tell" the right employees what to do to solve a particular challenge. There are rule-based management systems mainly based on ML. Standards for deceptively based programs are explained manually, while ML-controlled applications use models created with ML algorithms. Those patterns are recognized based on statistical tests; and they are tested, approved, and updated often.

Phase 5: automatic traffic control

Next to the opportunity for automatic control, there should always be an option for customers to direct smart city programs (for example, in the case of an emergency). User programs perform this task. Citizens can make use of user programs to connect to the city management platform to detect and manage IoT devices, and receive warnings and notifications. A smart traffic control solution, for example, identifies a visitor jam using global positioning system (GPS) data from driver's smartphones. The response sends out an automatic message to local drivers, requesting that they seek alternate routes.

The smart city area uses a tourist control system to discover over-congestion in real-time and to use tourist guidelines to reduce site visitors within high-traffic areas. Moreover, the smart city ensures that site visitors do not harm the environment and combine a visitor management response with a smart air tracking solution. The staff of a visitor center, using a computer application, can receive a warning about crowded conditions. To relieve and reroute traffic congestion, a command is sent to the robot detectives to control the alerts.

Phase 6: integrating multiple solutions

The IoT-based multiple solutions should be integrated, which means increasing not only the various senses, but, more importantly, the number of features.

### 1.3. Functions of Cloud Computing in IoT

IoT and cloud computing are two platforms that have proved to be beneficial in many ways. The majority of people are aware of IoT policies related to smart cities, smart homes, etc. IoT is key to integrating smart city's responses into business tools and paving the way for high-quality feedback in healthcare, transportation, logistics, energy, and many other fields. The cloud is not far behind. The benefits of cloud computing in IoT are numerous. In other words, IoT and cloud computing are extremely compatible and both endeavors to increase the efficiency of daily activities. While IoT integrates with smart cities, it produces large quantities of data. Cloud computing, on the other hand, paves the way for more experiences. From service options to the access of remote data, IoT and cloud computing together enhance integration. They provide accessible and cost-effective storage, but there are many areas where we can analyze the gap between IoT and cloud computing [6–10].

1. Cloud computing has made a considerable difference in the solutions of business services and individual applications. In addition, the intensity and strength of cloud response statistics allow data to be made available remotely. As a result, it has proven to be a solution to transferring information through network channels and hyperlinks delivered directly based on business preferences;
2. The cloud is an excellent IoT helper that solves the challenges driven by commercial business data. The cloud, as a technology, provides an active platform for developing critical applications for the better use of online data;
3. Velocity and scale: the two main cloud computing methods are an unparalleled combination, and IoT provides communication and mobility. Therefore, the capabilities of IoT and cloud computing are enhanced through combination. Other features prove that the cloud is important for IoT access;
4. Depending on the building infrastructure, with the widespread use of IoT devices a significant amount of time is required to maintain a large number of devices and to control over-speed. In this context, the cloud brings the benefit of a good service environment;
5. The cloud improves the security and privacy of IoT data. IoT devices are portable and, with the involvement of the cloud, they can integrate significant security measures, renovations, and discoveries. With robust authentication and encryption agreements, the cloud empowers customers by providing full security features;
6. The connection and presentation of cloud services for IoT devices. With plug-and-play cloud hosting services, considerable infrastructure is often required, which is expensive for organizations or individuals. With the combined power of IoT and

cloud computing, this investment in infrastructure is not required and any access restrictions for IoT and cloud service providers are removed;

7.  Advanced device connectivity: the cloud plays the role of a communication facilitator with its powerful IoT APIs. These APIs aid the pure connectivity of smart devices and also help in the conversation between intermediate tools;

8.  Cloud technology prevents companies from the necessity of infrastructure development and, at the same time, provides adequate resources;

9.  Cloud computing ensures business continuity, protecting against unexpected challenges that may arise throughout the process. Since the data is stored on separate servers, there is no risk of data loss, especially in particularly well-supported infrastructure;

10. Development within the IoT domain requires trouble-free secure responses. Therefore, cloud computing on IoT is the best solution. With cloud computing in place, IoT devices can use the power of remote statistical environments through applications. From a financial perspective, cloud computing on IoT is an excellent solution, as users successfully comply, and it saves considerably on future expenses. As a result, businesses may be able to utilize larger IoT systems. This reduces the access limit for high-level IoT-based organizations;

11. Cloud computing on IoT allows for seamless communication between IoT devices, enabling numerous strong API connections between connected devices and smart devices. In this way, cloud computing opens the way for the IoT explosion of connected technologies.

### 1.4. How Does the Cloud Allow IoT Applications?

Cloud-enabled IoT applications are growing and communicating across the network. The cloud enables service hosting, deployment, and the introduction of cloud-based IoT applications. Moreover, cloud computing is an appropriate Internet platform for storing and processing smart device data, such as connected cars, smart grids, smart cities, Wi-Fi, sensors, and actuator networks. The setup of network configurations can be conducted quickly and effectively. However, backend operations are performed using software, allowing rolling back, location monitoring, content labeling, and performance monitoring [11–15].

Additionally, cloud computing makes IoT systems robust. Using the integration of cloud and IoT, technologists can develop backups of devices and applications running in the cloud, increasing their tolerance to errors. In addition, they can be used to track data offline. Developers can also set up digital servers, run applications, and launch a database to help drive their IoT response.

### 1.5. What Are the Most Demanding Conditions Related to IoT and Cloud Computing?

Cloud computing accelerates the IoT explosion, and the integration of IoT and cloud computing can play a key role in the development of smart cities. However, IoT and cloud computing construction is complex, and the most demanding scenarios come from information generated at the network level [16–20]. The IoT cloud solution poses many challenges for users, including:

1.  Dealing with many records

With several millions of devices in the network, the IoT produces a considerable amount of information. In contrast, there is no straightforward or proven cloud management system for the handling of big data. This can put the full functionality of applications at risk.

2.  Communication processes

IoT and cloud computing includes device communication. This communication of devices to the machine occurs through a variety of processes.

3.   Sensitive networks

Sensors are the primary source of IoT data. The sensory community enables users to comprehend and respond to critical directions from the environment. However, processing a large number of sensory records is a considerable undertaking. While the cloud helps compile the data, it likewise prevents privacy and security issues.

4.   Cloud provider for IoT

Many businesses host their cloud platforms from secure locations for instant access to data. However, that may not be the most inexpensive solution. Therefore, a preferred cloud delivery service is a suitable response for organizations that share IoT responses. Currently, AWS and Microsoft Azure services are the leading cloud providers of IoT applications.

The rest of this paper is organized as follows: Section 2 presents the IoT and cloud convergence; Section 3 covers cloud-based IoT solutions; Section 4 presents cloud-based IoT applications for smart cities; Section 5 addresses the question of "why cloud-based IoT applications are essential for smart cities"; Section 6 discusses smart city applications; and Section 7 presents the conclusions.

## 2. IoT and Cloud Convergence

As IoT applications generate large quantities data and include multiple computational add-ons (e.g., real-time processing and analytics processes), integration with cloud computing infrastructure can be cost saving. Consider the following scenario as an excellent example. In a small to medium-sized enterprise, that manufactures a power control device used in smart homes and buildings, their ambitions for expansion may be unpleasantly and expensively achieved by spreading product details (e.g., sensors and WSN data) in the cloud. As small and medium-sized enterprises (SMEs) gain more extensive clientele and greater visibility for their product, they may collect and utilize an increasing amount of data. Furthermore, cloud integration enables SMEs to preserve and handle enormous data sets gathered from several sources [21–23].

A smart city can benefit from cloud-based building and system distribution. Intelligent power management applications, smart water controls, smart transportation controls, urban mobility, and other IoT packages are expected to be features of the smart city. Furthermore, they may provide higher volumes of data. The smart city can now handle these records and applications through estimating the cloud integration. Moreover, the cloud can assist in speeding up the expansion of the aforementioned packages and the deployment of recent ones, which have previously elicited substantial concerns over the provision of the necessary computer resources. A public cloud computing provider can expand the IoT ecosystem by granting third-party access to its infrastructure, allowing them to combine IoT data and computer resources operating on IoT devices. The company can provide IoT data for access and service. This shows the applicability and desire for IoT infrastructure and cloud computing modification. Integration has always been problematic, however, owing to the conflicting IoT and cloud architecture. IoT devices tend to be regionally constrained, with limited support, expensive estimating (depending on upgrade/shipping cost), and frequently unstable (according to resources and access). On the other hand, cloud computing resources are typically located in a reasonable and efficient place that provides rapidity and flexibility. Sensors and devices are established before integrating data and their offerings into the cloud, allowing them to distribute across any cloud resources and reducing inconsistencies.

Furthermore, service implementation and sensor acquisition are placed in the cloud so that services and sensors are available in real-time. IoT and cloud integration can transfer sensory and WSN information to cloud. This widespread infrastructure was one of the earliest innovations (widely used for radiation detection and radiation maps during earthquakes in Japan). There are, however, dozens of well-known clouds, including ThingsWorx, ThingSpeak, cloud-sensor, and real-time cloud services. Consumers that wish to save IoT packages in the cloud can pay as they go with these public cloud providers. Most

of the providers include advanced developer tools that enhance cloud systems, making them akin to IoT services in the cloud. Additionally, cloud computing infrastructure, IoT/cloud infrastructure, and associated services might be designated as follows:

### 2.1. Infrastructure as a Service (IaaS)

IoT/clouds allow users to connect to sensors and actuators in the cloud. IaaS is a significant computation, which stores and communicates a solution upon request. It provides a cloud computing service. IaaS offers IoT management to manage objects as a prerequisite in the supply of appropriate services [24–27].

### 2.2. Platform-as-a-Service (PaaS)

The IoT/cloud public infrastructure described the high-performance PaaS model for IoT/cloud services. PaaS is a full cloud-based development and deployment architecture, including capabilities to provide services that range from the simplest cloud-based services to complex, cloud-capable businesses.

### 2.3. Software-as-a-Service (SaaS)

SaaS products are those that allow users to obtain complete software packages based specially on the cloud and IoT. The SaaS packages are similar to standard cloud-based packages using IoT sensors and devices.

The assertion is that SaaS IoT packages are typically developed on top of PaaS infrastructure and enable business models that are dependent on IoT software and services. It gives a broad understanding of IoT and cloud interactions and why they are so significant and advantageous. At present, a rising number of IoT devices are cloud-based, allowing users to benefit from their overall performance, business expertise, and the payment features they provide. The benefits of the IoT cloud, by assuring the interoperability of IoT data and contributions within the cloud, are optimized, which is why they enable high records for analytic purposes in regions involving smart energy, smart transportation, smart cities, and communications. Moreover, IoT components with IoT-based wearable computing can profit from IoT/cloud integration.

The IoT is a mechanism for connecting computer devices, machine tools, and virtual objects, animals, or people that were given indications and the power to modify records on a network without the need for human or computer contact. The IoT umbrella encompasses anything produced by humans or devices that can be specified by an I.P. address and possess the ability to send data over a network. With the advancement of information technology, the IoT has expanded. IoT devices enable communication between sensors, with billions of connected devices likely to become part of human lives in the future

Many businesses around the city, including the agricultural sector, health care, energy, transportation, and building management, were nearly completely taken over by IoT. Experts and active developers are attempting to determine other ways to connect to the IoT via cloud network. IoT applications are being developed, which are an extraordinary approach to aid future development. People no longer gain from the increased connectedness of devices, but the socially relevant devices they collect from IoT network. Machines can provide helpful information about performance and appearance in the field to the communication enabled by cloud solutions. Related devices are not limited to certifying businesses' devices, but they can also move away from personal devices for everyone via network cloud solutions. Through closed storage, processing power, energy, and other quick connections, the timing of things is controlled with the help of reality. Due to the wide variety of sensors, and the number of records they produce, the collection, storage, and efficiency of IoT remains challenging.

IoT also links devices and people while generating massive amounts of data. Due to sophisticated systems, typical connection agreements, and legacy application compliance, gaining access to information through enterprises can be a difficult task. IoT infrastructure (e.g., sensors, WSN, and RFID) is either unique or ubiquitous, and resources to build and

convey access are restricted and often expensive. IoT devices are frequently afflicted by deficient processing and storage resources and a limited budget due to their productivity. Cloud computing provides limitless storage and is energy efficient. The cloud connects users to information and resources via an Internet-based link. Cloud infrastructure is a self-contained or ubiquitous region (resources that may be accessed from anywhere) that enables simple access to lower-cost resources. Virtualization in the cloud is the result of the resource base resources' autonomous effort. The IoT-cloud connection is a way to experience the resources and is constantly available during cloud computing. The demand for IoT systems for cloud compression, availability, and performance promotes the integration of IoT and cloud computing technology. This connection allows for the storing and processing of accumulated facts, identical data in different contributions, the integration of points from multiple devices and users, and user mobility. Several attempts were made to integrate IoT and cloud technology into the research network and the commercial community. The capacity to transfer data to the cloud because of this combination of power is extraordinary in terms of working, managing systems, monitoring, and controlling the distribution of data. For IoT packages, the cloud can make use of reliable restoration and processing equipment and retrieval packages. Large-scale IoT systems are inherently insecure. Large IoT systems contain a diverse set of diffuse sensors that create data and must be addressed. The cloud offers an oversized output service and can use IoT technologies to build a complicated system. Furthermore, many IoT services can benefit from a delivery system that focuses on establishing and delivering IoT systems and is primarily based on cloud infrastructure.

A platform for the IoT is primarily based in the cloud, with the capability to design, deploy, run, and manage networks of cloud-based IoT devices. This depicts the initial capabilities of the cloud-based IoT platform and architecture, in addition to their interactions with the three cloud computing fashions (i.e., Iaas, PaaS, and SaaS). All IoT devices will connect with a cloud-based ubiquitous resource pool. Devices can access without difficulty, accumulate, systematize, visualize, archive, proportion, and seek full-size volumes of sensor information from many programs using this platform.

The cloud's computational and storage resources may be used to produce, analyze, and save sensor information. Moreover, a cloud-based IoT platform permits customers and programs access to percentage sensor data underneath flexible usage situations, permitting sensor devices to meet specialized processing responsibilities. This platform is an extended-time period cloud computing solution for sensor control that includes sensor devices as a supply to the customers. It provides sensor monitoring and manipulates offerings to customers through a web browser. Furthermore, the cloud simplifies data throughout IoT information collecting and processing additives, considering simple setup and integration of latest issues, while retaining low deployment charges and complex data processing. Clients can execute any application on cloud hardware with the use of this platform's cloud infrastructure.

The platform streamlines application improvement, eliminates the need for infrastructure, makes problems less challenging to manipulate, and lowers refurbishment charges. It provides clients with unique tool control abilities, direct communication with devices, storage to accumulate data from issues, and event transition. The cloud's computing and service belongings may be used to save, process, and analyze a significant amount of sensor data. The developer suite is fixed on complex and rapid cloud carrier devices for developing IoT applications. This technology encompasses open service software programming interfaces, providing developers with high-degree improvement and deployment capabilities. Subscription management, network coordination, subjects' connection, matters discovery, statistics intelligence, and things composition are all part of the system, a package deal of cloud offerings that help with the deployment and specialization of processing services.

IoT devices are commonly grouped into IoT networks in cloud-based IoT structures, including a domestic community. Those networks are connected to the cloud through

a dedicated gateway, generally a home router or a mobile phone. The committed gate-manner forwards the detected information from the networks to the cloud. The cloud continuously retains data and ensures it is on hand to contribution to applications. The provider can provide specific cloud services with permission to access and manage the information through cloud processing belongings. The cloud serves as an intermediary layer among issues and IoT applications, concealing all the implementation's complexities and functionalities. This platform can improve destiny software since data collection, and records transmission will offer a new solution. The design of the cloud-based, completely IoT platform aims to maximize the supply of data and services. Figure 2 depicts the cloud-based IoT platform connected through the cloud-based IoT applications. The cloud applications can be stored and visualized so that the customer may also access, monitor, and control from everywhere and at any time using a web browser or application.

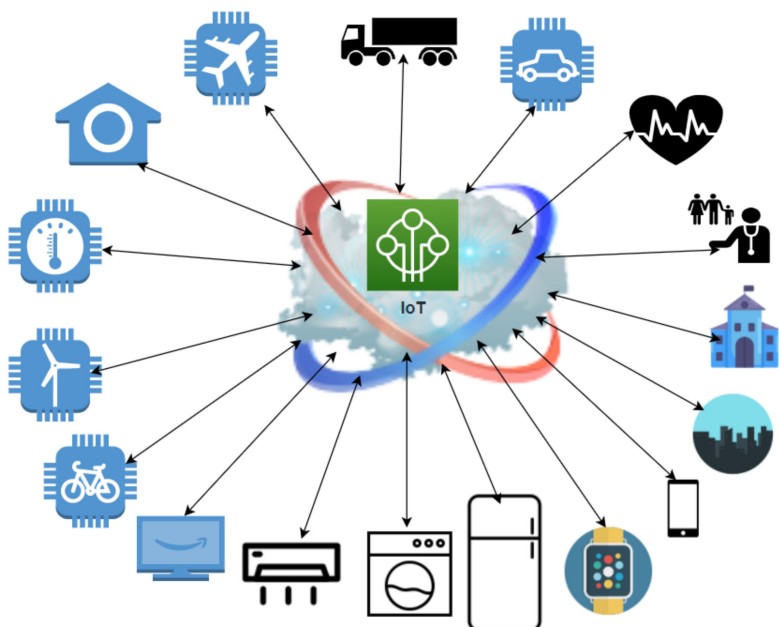

**Figure 2.** Cloud-IoT applications main areas.

## 3. Cloud-Based IoT Solutions

Many organizations recommend cloud-based systems for IoT devices. Provided below is a list of the companies and corporations that offer cloud-based IoT services. We can select one from the IoT system to obtain the required cloud platform for data collection and testing. With more than one IoT-based application, many statistics are created with the help of IoT devices.

### 3.1. Xively

A cloud-based solution that allows multiple teams to fix IoT-related problems without difficulties. The offer comes from the agency's trademark (linked product control), which assists in keeping records secure and interactive [27].

### 3.2. ThingSpeak

ThingSpeak is one of the most notable IoT-related names and was responsible for the conversion of many products within the IoT. It can expand the task in three easy steps: collection, analysis, and development. ThingSpeak features multiple offers, such as information recognition and analytics tools. With growing customer issues and building a reputation, ThingSpeak is a platform that holds valuable IoT data securely. Collected data can be stored on private or public cloud networks [28].

### 3.3. Plotly

Plotly is an open-source API, which is mainly used to transmit data from IoT devices, such as Raspberry Pi and Arduino. Services, such as formal or systematic cloud, can be used to streamline data. Plotly is more determined in terms of significant facts and its analytical strategies, making use of them to set displays and facts differently from your devices. It is one of the most popular data analysis and recognition tools among IoT professionals [29].

### 3.4. Exosite

Exosite is a prominent IoT framework for the building of linked things, solutions, and companies. It is fully-fledged organization, which offers cloud-based offerings to expand, launch, and incorporate the most advanced ideas into your applications. It provides a number of monitoring and analysis statistics, and is a significant business venture dedicated to development and innovation. It likewise includes the overall package from developing to enhancing existing sales skills. With automatic updates and installation services, this is an excellent platform for industrial applications. This platform is easy to use for builders of cloud-connected applications [30].

### 3.5. Grove Stream

One of the most widely used online systems, this platform is used to attach thousands of flows of large amounts of data via cloud computing. You can visualize, explore, store, and share data. It provides a sophisticated set of tools for users to investigate sensory information that help IoT applications. It also offers an extensive network of cloud statistics in addition to the data that drive the service and server [31].

### 3.6. Axeda

Axeda offers cloud-based contributions to build and strengthen IoT operations. This cloud platform is used for many tasks, such as building, converting data from sensors to information, and navigating IoT systems. In addition, it introduces awareness of the functions associated with IoT and M2M. The cloud device provides transforming strategies to manage advanced, technique-based cloud IoT applications [32].

### 3.7. ThingWorx

ThingWorx is a critical product within the IoT system. It provides an inexpensive and robust solution for cloud delivery to IoT. It has four crucial built-in functions for analyzing, constructing, managing, and gathering IoT data. With many plug-ins and extensions, this is considered one of the most widely used applications for connecting many devices to the cloud [33].

### 3.8. Yaler

Yaler is used to transmit large quantities of data among devices, an important feature that is used in the electronics industry and the Internet of devices. With the introduction of the new market, it has accelerated its use and operations into the IoT cloud. It offers cloud-based connectivity service as one of its top offerings [34].

## 4. Cloud-Based IoT Applications for Smart Cities

Table 2 shows the list of cloud-based IoT applications for smart cities.

**Table 2.** Cloud-based IoT applications and their roles in smart cities.

| Applications | Ref. | IoT Based Application | Use Cloud Computing | Future Perspectives | Roles in Smart Cities |
|---|---|---|---|---|---|
| Bosch IoT Suite | [35] | √ | √ | √ | 1. Fast project success<br>2. Maximum security<br>3. Future-proof investment<br>4. End-to-end service solution<br>5. Flexible deployment |
| ABB Robotics | [36] | √ | √ | √ | 1. Robotics<br>2. Machine automation<br>3. Digital services<br>4. Providing innovative solutions for a diverse range of industries |
| Airbus | [37] | √ | √ | √ | 1. Design, manufacture, and deliver industry-leading commercial aircraft, helicopters, military transports, satellites, and launch vehicles<br>2. Providing data services, navigation, secure communications, and urban mobility. |
| Amazon Warehouse | [38] | √ | √ | √ | 1. E-commerce<br>2. Cloud computing<br>3. Digital streaming<br>4. Artificial intelligence |
| Boeing | [39] | √ | √ | √ | 1. Commercial and military aircraft<br>2. Satellites<br>3. Weapons<br>4. Electronic and defense systems<br>5. Launch systems<br>6. Advanced information and communication systems, and performance-based logistics and training |
| Caterpillar | [40] | √ | √ | √ | 1. Data-driven transformations<br>2. Intelligence platform<br>3. Shipboard sensors monitor everything from generators to engines, GPS, air conditioning systems, and fuel meters |
| Fanuc | [41] | √ | √ | √ | 1. Manufactures all products in highly automated factories<br>2. Factories are highly automated, with all devices being connected to a network<br>3. Each process is automated with robots and connected with an automatic transport system |
| Gehring | [42] | √ | √ | √ | 1. Honing technology<br>2. Supplying cutting-edge surface finish technology solutions for internal combustion engines, gears, and numerous other industrial applications |
| Hitachi | [43] | √ | √ | √ | 1. IoT-ready industrial controller<br>2. H.X. Series Hybrid Model<br>3. Executing control functions, such as sequence control and motion control<br>4. Execute information system communication |

**Table 2.** *Cont.*

| Applications | Ref. | IoT Based Application | Use Cloud Computing | Future Perspectives | Roles in Smart Cities |
|---|---|---|---|---|---|
| John Deere | [44] | √ | √ | √ | 1. Electrification<br>2. eAutoPowr transmission<br>3. Large spraying drone (VoloDrone)<br>4. Autonomy through automation<br>5. Artificial Intelligence |
| Kaeser kompressoren | [45] | √ | √ | √ | 1. Compressor system design to professional air service<br>2. Exceptional energy balance<br>3. Time-saving maintenance and operation<br>4. Exceptional material quality and durability |
| Komatsu | [46] | √ | √ | √ | 1. Construction<br>2. Demolition, waste, and recycling<br>3. Mining<br>4. Agriculture and livestock<br>5. Logistics<br>6. Industrial machinery |
| Kuka | [47] | √ | √ | √ | 1. Robot systems<br>2. Automated guided vehicle systems<br>3. Mobility<br>4. Process Technologies |
| Maersk | [48] | √ | √ | √ | 1. Global air freight transportation<br>2. Most time-efficient freight solution for many destinations around the world<br>3. Inventory costs reduction while improving flexibility<br>4. Highly reliable arrival and departure times |
| Magna Steyr | [49] | √ | √ | √ | 1. Ideal automotive contract manufacturer<br>2. Produce vehicles with conventional, hybrid, and electric powertrains |
| Bluescope | [50] | √ | √ | √ | 1. Provider of innovative steel materials, products, systems, and technologies |
| Real-time Innovation | [51] | √ | √ | √ | 1. Largest software framework provider for autonomous systems<br>2. Enabled comprehensive connectivity |
| Rio Tinto | [52] | √ | √ | √ | 1. Produce iron ore for steel, aluminum for cars and smartphones, copper for wind turbines, etc. |
| The Shell | [53] | √ | √ | √ | 1. Shell is a global group of energy and petrochemical companies that aims to meet the world's growing need for more and cleaner energy solutions in ways that are economically, environmentally, and socially responsible |
| Stanley Black & Decker | [54] | √ | √ | √ | 1. Outstanding performance and ceaseless innovation<br>2. Use industrial tools that build and rebuild infrastructure<br>3. Provide the security services and solutions |

### 4.1. Bosch IoT Suite

In 2015, Bosch introduced the first toolbox in the cloud for IoT developers. This was an important invention, and supported future-oriented products and the design of emerging revolutionary commercial strategies. IoT is the essential technique for improving practices, performance improvement, and continuous improvement. IoT allows businesses to learn consumer demands, enhance flying operations, and instantaneously introduce unique characteristics [35].

### 4.2. ABB Robotics

The IIoT is one of the most visible to undertake the idea of protection and the usage of connected sensors for robots. ABB Robotics is a global leader in robotics, industrial automation, and cloud solutions, offering cutting-edge products to a wide range of sectors, including manufacturing, telecommunications, and transportation. ABB Robotics engages over 11,000 professionals in 53 nations and has sold over 500,000 robotic systems, making it one of the world's most extensive robotics and industrial intelligence companies [36].

### 4.3. Airbus

Airbus is a company that provides IoT solutions all over the globe. Regarding world-wide connection, Airbus uses Astrocast nanosatellites to deploy its IoT technologies and connectivity network. Inside a centralized design, the IoT application has a communication link. It relies on a dependable infrastructure that can accommodate thousands of users utilizing sensors and functioning on the same radio signals [37].

### 4.4. Amazon Warehouse

The supermarket that sells online is not generally known as an IIoT enterprise, but the organization is the inventor of things approximately inventory and inventory planning. Amazon looks at the limits of automation and human interaction, and the enterprise's coverage of the use of delivery drones has obtained considerable attention [38].

### 4.5. Boeing: The Usage of IoT to Force Manufacturing Performance

Aviation pioneer William Boeing stated that it "behooves no one to dismiss any novel idea with the statement, it can't be done". The worldwide airline founded on Boeing is manifestly in agreement with that ethos. The enterprise has made splendid strides in reusing its commercial enterprise. Boeing and its solutions have aggressively used IoT time to force operations among factories and convoy chains. Smart technologies that enable travelers to engage with the aircraft as never before, such as devices that speak to smart toilets or smart lighting handled by the networks, might be introduced into the commercial flight cabin environment [39].

### 4.6. Caterpillar: IIoT Pioneer

Caterpillar is using the Internet of Things to boost production. Heavy equipment has long been a pioneer of IoT initiatives. Throughout aggregate, 560,000 Caterpillar trucks are networked across the globe. Furthermore, the firm has developed a collection of software and analytics tools and APIs to assist it and its clients in processing, analyzing, and storing information. Whereas the firm has an in-house analytics department, it has also developed a community of partnerships to give customers various alternatives for these objectives. Notably, the firm has formed a new collaboration with Zuora to provide customers with cloud-based solutions for managing and analyzing subscription facilities [40].

### 4.7. Fanuc: Supports Decreasing the Downtime in Factories

This maker of robots is determined to reduce the downtime in industrial facilities. Through the usage of sensors in its robots, alongside cloud-primarily based analytics, the organization can determine when the failure of a robotic device or system is imminent. Fanuc strives to reduce downtime in all of its factories across the globe. It offers continuous

servicing to its products for as long as they are utilized by consumers, with more than 260 service centers in 108 nations, showing an adherence to the philosophy of "Service First." [41].

### 4.8. Gehring: Pioneer in the Production

Gehring generation, a 91-year-old manufacturer of metallic sprucing machines, started early to adopt IIoT manufacturing. At present, the organization empowers its customers to obtain details of the way Gehring machines paint prior to receiving an order. It does so via virtual technology, illuminating real-time realities from modern machines to ensure it meets customer needs. Gehring makes use of real-time cloud-based monitoring to lessen downtime and improve its productiveness through its related production centers, visualizing and controlling data from devices within the cloud [42].

### 4.9. Hitachi: Established IIoT Manner

Hitachi is a Japanese company that adheres to diverse change institutions to integrate its software and operations. The H.X. Family Combo Models is a revolutionary IoT-ready industrial controller from Hitachi. This controller could perform computer network communications and software platforms tailored to data processing and functionalities, suchas sequence control and motion control, without impacting control functions [43].

### 4.10. John Deere: Future of Farming

Changing climate conditions is only one of the numerous problems that farming faces. John Deere is devoting a significant amount of effort towards addressing these issues. Electrification, automation to autonomy, and artificial intelligence are three key innovations that will shape the future. Due to Washington's position in 2015, Google no longer leads the revolution in self-driving automobiles and, instead, John Deere does. The business is smartly designed to pioneer using GPS [44].

### 4.11. Kaeser Kompressoren

In 1919, the German producer of air pumps, air dryers, and filters has integrated digital communications into its merchandise. Kaeser Kompressoren is a leading provider of air compressor solutions across the globe, with around 7000 people working for the firm [45].

### 4.12. Komatsu

The Japanese heavy machine producer has plenty of the latest IIoT substances. In 2011, it spent the period linked to its Japanese manufacturing facilities. Komatsu has related all its robots to its key production centers and network, allowing managers to keep an eye on global operations in real-time. The organization is a mining revolution. Its big pickup vehicles will be visible at the future Rio Tinto mine in Australia [46].

### 4.13. Kuka: Robots

German robotics professional, Kuka, has an IoT system that reaches many enterprises. Kuka makes robotic systems and factory industrial Internet of Things. The Kuka consultation method is built on a flexible training philosophy that constantly focuses on providing value addition [47].

### 4.14. Maersk: Global Air Freight Transportation

Maersk is well-known for its industrialized cargo shipping services, but the firm is expanding beyond that. Maersk is becoming an end-to-end supply chain logistics company by integrating solutions. The firm utilizes Microsoft Azure IoT technology to detect and control 380,000 refrigerated containers as they travel across the globe. Customers will always know wherever their goods are, and environmental factors may be changed to ensure that the food and medication from one side of the global reach the other in pristine

condition. It is confident in Azure's IoT capabilities because of the platform's development and adaptability [48].

### 4.15. Magna Steyr: Smart Automotive Production

Austrian car producer Magna Steyr is a pioneer within the field of smart industry. It has 161,000 employees, working on automobile parts and automotive components, automatically ordering cashback although emerging technologies. Magna is also experimenting with using "smart packaging" and improving it through IoT [49].

### 4.16. BlueScope

As per Andrew Spence, BlueScope Constructing Products' regional director of operations and production, smart, connected devices enable the firm to improve current systems and develop entirely new alternatives. Process parameters are pressure, temp, flows, ongoing, tier, mobilization, and acid content. For example, according to Spence, improvements to the production process and producing information that could be used in designs to assist more advanced and accurate control strategies can now be easily evaluated through the Internet of Things. The Internet of Things is also helping to improve the safety and health of workers [50].

### 4.17. Real-Time Innovation (RTI): The Largest Software Framework Provider for Autonomous Systems

The leading software framework firm for autonomous systems is Real-Time Innovations (RTI). It is the most widely used paradigm for creating smart decentralized systems around the world. Its context is unique in that it immediately distributes information, linking A.I. techniques to real networking of devices to develop autonomous systems. It offers an established product portfolio that allows hundreds of applications to securely share data in real-time and function as one integrated solution in the race to serve clients who are changing the world into a better place [51].

### 4.18. Rio Tinto

Rio Tinto generates elements that are necessary for human growth. The British/Australian mining conglomerate has unveiled a new computerized program in Pilbara, western Australia. Non-motorized trucks and trains pull steel far away from mining sites simultaneously as a faraway operator controls drills from an available console. Non-pilot vessels may be appropriately terminated. The organization has a Perth-based facility that connects to its mines, similarly to rail and port operations, wherein engineers, analysts, planners, and experts remotely direct mining operations [52].

### 4.19. The Shell

Shell was named the most significant oil and gasoline exchange employer in a Rigzone in 2016. Shell's senior chairman for technology officer, Yuri Sebregts, stated, "The new possibilities in dealing with data over the last several years have unlocked great potential in all parts of what we do in the firm", and "Right now, this will assist us in scaling ideas we've been going to develop". Shell is also working with Microsoft experts to enhance horizontal drilling using A.I. and machine intelligence. By switching from traditional drilling wells to lengthy, horizontal drilling, the oil and gas sector has realized considerable cost savings, decreased its footprint, and discovered new oil and gas resources onshore [53].

### 4.20. Stanley Black & Decker

Stanley Black & Decker delivers the equipment and creative solutions for the constructors and adventurers, creators and explorers, and those influencing and remaking the world through hard work and imagination. It bands together and brings the best in the world to build realistic, relevant goods and solutions that make lives more accessible, while also enabling individuals to have smarter, healthier, and more rewarding jobs. This

achievement was fueled by quality and creativity, and through the understanding that there is more that they can do for the world. by providing value to customers, coworkers, and societies [54].

## 5. Why Cloud-Based IoT Applications Are Essential for Smart Cities

Cities have transitioned to IoT technologies and communication technologies for a variety of reasons.

- The IoT systems enable sensors to detect data to manage appliance consumption, potentially resulting in significant cost savings;
- Since installing and maintaining IoT applications is more accessible, the cost is a significant consideration when determining whether to go physically or online. Furthermore, the prices are decreasing, and communications' durability and power output allow for new circumstances that were previously not possible;
- Efficiency is one of the most significant considerations [55]. Service providers must physically go to the web page to examine and execute communications infrastructure for the most stressful solutions;
- Wireless communication provides monitoring and control of IoT transmission through various analyses. This allows administrators to upgrade firmware and apply security solutions to all completed plans and get automatic alerts in the event of an issue;
- Reduced assistance is frequently the cause, as it should be, especially in operating situations and when smart road lighting and tracking equipment are repaired.

Table 3 represents the previous studies on smart city solutions.

**Table 3.** Smart Cities Solution (previous studies).

| | | | |
|---|---|---|---|
| Lea et al. | [55] | 2014 | City hub: A cloud-based IoT platform for smart cities |
| Sikder et al. | [56] | 2018 | IoT-enabled smart lighting systems for smart cities |
| Ding et al. | [57] | 2018 | Intelligent data transportation in smart cities: A spectrum-aware approach |
| Ramos et al. | [58] | 2020 | Smart water management towards future water sustainable networks |
| Chung et al. | [59] | 2021 | Smart Tourism Cities' Competitiveness Index: A Conceptual Model |
| Biyik et al. | [60] | 2021 | Smart Parking Systems: Reviewing the Literature, Architecture and Ways Forward |
| Miyasawa et al. | [61] | 2021 | Spatial demand forecasting based on smart meter data for improving local energy self-sufficiency in smart cities |
| Khalifeh et al. | [62] | 2021 | Wireless Sensor Networks for Smart Cities: Network Design, Implementation and Performance Evaluation |
| McCurdy et al. | [63] | 2021 | Waste Management in Smart Cities: A Survey on Public Perception and the Implications for Service Level Agreements |
| Chatterjee et al. | [64] | 2021 | Smart Cities and Their Quality of Life: An Interdisciplinary Perspective |
| Múnera et al. | [65] | 2021 | IoT-based air quality monitoring systems for smart cities: A systematic mapping study |
| P Kasznar et al. | [66] | 2021 | Multiple Dimensions of Smart Cities' Infrastructure: A Review |

## 6. Smart City Applications

"Smart cities" are a collection of enterprises that include city lighting, traffic, wastewater management, emergency services, tourism management, and so forth. Inventive new city occupations are likely to become more widely adopted and technology focused based on the needs of specific use cases. Figure 3 shows the smart cities applications.

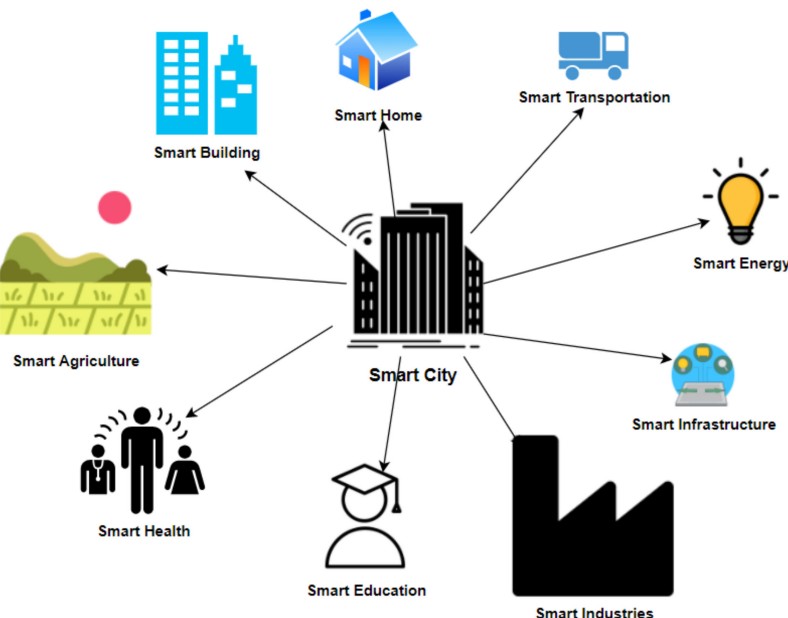

**Figure 3.** Smart cities applications.

### 6.1. Lighting Systems in Smart Cities

Light sources are one of the most ubiquitous IoT applications for smart cities, with several governments currently relying on IoT to save money and energy. The system includes a ruggedized Digi wr44r router class, which provides connectivity and authentication for transferring numerous device nodes to a smart pole. The smart lighting may be used for a range of tasks, as follows:

1. Controls for lighting;
2. Cameras for surveillance;
3. Natural perceptions;
4. Electronic billboards;
5. Electric vehicle charging stations;
6. Access to wireless technology.

The use of IoT in smart houses makes street light maintenance and management practical and cost-effective. The lighting can be synchronized by equipping streetlights with sensors and linking them to a cloud management service [56]. Smart lighting systems monitor light, people, and vehicle movement, then integrate it with old and contextual data (e.g., unique functions, public delivery system, time and year, etc.) and analyze it to enhance the lighting schedule. When pedestrians cross a road, the lights around the crossing could be turned on, when a bus is about to arrive at the bus stop, the streetlights could be brightened, and so on.

### 6.2. Transportation

Transportation infrastructure is another rapidly expanding component of smart city applications. Transportation businesses and smart cities stand to gain considerably in cost savings, security, route management, and advanced passenger experience. While many communities have seen a reduction in shipping in recent years with the advent of buses and trains and wireless passenger links, many are now experiencing additional improvements. They help the smart, urban transit organization for fast-moving authorities, as they operate over 330 biodiesel buses and power buses in all of Michigan's Macomb, Oakland, and Wayne cities. The smart transportation system's IoT response contains several functions, all of which are powered by the Digi wr44 r router:

1.  The vehicle circuit system and a wireless connection are between the motors and the smarts dispatch center. This increases concerns regarding the transition from the existing analog network to IP-based voice-to-voice communication;
2.  Collection of comfortable fares and mobile ticketing are via wr44 r state wall firewall, IPsec VPN, social isolation, and verification;
3.  Intermediate communication and passenger controlled;
4.  With online service, we can track and maintain our groups and devices, including crowd updates and vehicle monitoring—these enhancements aid transportation employees, couriers, and passengers in feeling safer in their communication and development [57].

Furthermore, as the number of green automobiles increases, smart cities will experience a fresher atmosphere. In addition to the modernization of vehicles and suburban residents who travel by car, the goal is to reduce traffic congestion and air pollution.

*6.3. Water Management*

Water management software in smart cities is used for various purposes, including wastewater solutions, water tracking, and environmental restoration projects [58]. IoT packages are becoming more common in locations such as state-owned companies and nearby municipalities. It improves access to aging infrastructure, increases efficiency, improves visibility of remote tanks and water management plans, and lowers the cost of tracking and assisting their facilities. The gateway connects to a network of services that can help with various issues, such as tank pressure and water levels. Digi far-flung is a remote-control solution for testing the components of an IoT distribution tool, which can also integrate IoT devices and systems into modules and sensors. U.S. Water, which delivers water treatment services to commercial customers across the United States and Canada, evolved out of a response from Digi to establish a regional, remote tracking and management solution with their cutting-edge technology.

*6.4. Smart Tourism*

Finding ways to boost site traffic is one of the most challenging difficulties that big cities face. For example, Los Angeles is one of the world's busiest cities, and it has devised a smart shipping strategy to manage tourists. Sensors embedded in the pavement feed real-time traffic crash updates to a critical traffic control platform, analyzing the data and adjusting the site visitor lights to traffic. Simultaneously, past data is used to predict where traffic will travel, with none of these tactics requiring human intervention. Smart cities ensure that their citizens move as accurately and efficiently as possible from one place to another. Municipalities imposed IoT development and compelled visitor reactions to the smart site to achieve this goal. In addition to collecting GPS data on smartphones, smart traffic solutions use sensors to help drivers determine cars' range, location, and speed [59]. Simultaneously, intelligent visitor lights connected to a cloud control platform allow for the measurement of time and management of the lights, based on the present status of the visitors, to avoid traffic congestion. Furthermore, by analyzing past data, intelligent traffic management responses can predict where people will cross and take precautions to prevent power outages.

For example, being one of the world's most popular tourist locations, they employed tourist responses to help regulate traffic flow. On a robust guest management platform, street-surface sensors and closed-circuit cameras provide real-time information virtually to traffic flow. The platform analyzes data and sends alerts to the customers about traffic congestion and misuse of road signs through applications.

*6.5. Smart Parking*

Cities also employ smart parking solutions that detect when a vehicle has departed from a parking space. Sensors embedded in the ground notify the driver of available parking spaces via a smartphone application. Smart parking is an authenticity, requiring

no specialized infrastructure or significant expenditure. Smart parking responses check if parking spaces are available and construct a real-time parking map using GPS data from drivers' smartphones (or road level sensors embedded in the ground in parking lots) [60]. Drivers are told when the nearest parking space opened and, instead of relying on memory, they can make use of a map on their phone to find a parking space. IoT sensors can be utilized to send messages to the connected devices. Public transportation operators can use this information to improve visiting data, resulting in increased safety and punctuality.

Many train companies in London are waiting for passenger cars to be loaded for journeys inside and outside. The data are gathered from ticket sales, motion sensors, and CCTV cameras located near the platform. Train operators expect each car to load with people and urge customers to disperse from the area when a train arrives at a station to enhance loading. Train operators prevent train delays by boosting energy usage. Citizens in smart communities can save money by allowing them to handle extra resources at home.

### 6.6. Smart Meters

Cities can give citizens the most crucial linkages to service delivery structures through the smart meter community. Smartly connected meters can now transfer data to general applications over the network in real-time, giving it accurate meter readings [61]. With the cooperation of the entire group, the smart meter allows service organizations to charge the quantity of water energy, and fuel used more accurately.

Service organizations can be more visible and observe how their clients use power and water through the smart meter service. Resource organizations can use a smart meter service to show real-time calls and redirect resources as needed or encourage consumers to use less energy or water during shortages.

### 6.7. Smart Remote

Smart city solutions based on the IoT can also give inhabitants application management services. These contributions enable citizens to utilize their smart remote on, for example, their television and air conditioning to take advantage of their remote capabilities [62]. For example, a homeowner may switch off heating in his home using a mobile phone. Additionally, in an emergency (e.g., water leaks), utility companies can notify homeowners and send experts to remedy these issues.

### 6.8. Waste Management

Waste management solutions help to increase the efficiency of the waste chain and reduce operating costs, while, at the same time, dealing with any environmental concerns associated with an inefficient waste chain. In these responses, the waste container receives a stage sensor; while reaching the boundary, the truck driver's management platform gets a notification by their phone. The message helps them to avoid empty drains by performing the related task. Many open garbage collection operators can follow these procedures. IoT-powered city-based responses help increase waste collection schedules with the help of waste tracing and the introduction of methodology and performance analysis [63].

Each waste field receives a sensor that collects records about the level of waste in the area. The waste management solution detects sensor data, evaluates it, and sends a notification to the truck's mobile application. Similarly, the truck driver pours out the entire container to empty it. IoT smart city solutions in the surrounding region allow tracking of the crucial factors required for a healthy environment. A major city, for example, may incorporate a sensory community across the water grid and bring them together on a cloud management platform to reveal the most significant waste.

### 6.9. Social Security

IoT-based smart city technology provides real-time tracking, analytics, and alternatives for increasing social security. Public security systems can predict the power of crime by combining statistics from sensors and CCTV cameras provided to the city with data from

social media feeds and readings. The police would be able to dissuade or punish criminals as a result of these social security applications. The solution is to use networked devices in the smart city. For example, in the case of a crime, the device information is sent to a cloud platform, the data is analyzed, and the criminal is identified. The platform calculates the time and the distance between the gun and the mobile phone that reported a gunshot. The cloud software can then alert police with a mobile application [64].

### 6.10. Air Control Platform

Smart cities are also valuable tools for detecting and forecasting pollution in real-time. Cities can get to the source of their emissions problems and consider strategic approaches to reduce air pollution. Monitoring the amount of greenhouse gases in the air is essential; regulatory systems follow the rules and can be used to, for example, take control of tourists' local flights. Before that, there may be a need to ensure that visitor changes do not cause accidents in other areas. This is possible because of the combination of the way visitors control the air quality control system [65].

### 6.11. Smart Infrastructure

Building infrastructure must be planned carefully and effectively. Virtual technology is becoming increasingly important for cities to maintain growth conditions [66]. Cities should invest in electric motors and self-propelled vehicles [67] to cut carbon dioxide emissions. Smart technology is being leveraged to create energy-efficient and environmentally friendly infrastructure. For example, smart lighting provides light while someone passes through a smart lighting area, reducing energy expenditure [68,69]. Artificial intelligence could perform a key role in enabling wireless connectivity to the IoT infrastructure [70]. Smart cities will improve individuals' lives and can lead to a new age of efficient and data-driven decision-making, ranging from enhancing transport flows and allowing interconnected and affordable services, to wireless connections, mobile edge computing, and the IoT [71,72]. A tracking system for people, for example, designed to track children or the elderly in crowded environments using mobile applications for smart cities is discussed in [73]. The data processing in smart cities using blockchain-based big data integrity service is discussed in [74].

## 7. Conclusions

Accurate information could be accessed, analyzed, and controlled by cloud-based enabling technologies to assist experts, businesses, and people in making smarter policies to enhance the standard of peoples' life. People interact in smart city environments using their mobile devices through linked vehicles and smart homes. When devices and information are connected to a city's physical systems and facilities, expenses may be reduced and efficiency improved. Through the assistance of the Internet of Things, cities could enhance resource transmission, expedite garbage collection, reduce accidents, and remove pollutants. The author explored and discussed the cloud-based IoT applications and their roles in smart cities in this paper. The author also covered IoT and cloud convergence, cloud-based IoT solutions, and cloud-based IoT applications for smart cities. More applications can be discovered, and their importance in smart cities discussed, in future research.

**Funding:** This research received no external funding.

**Institutional Review Board Statement:** Not applicable.

**Informed Consent Statement:** Not applicable.

**Data Availability Statement:** Not applicable.

**Conflicts of Interest:** The authors declare no conflict of interest.

## Abbreviations

The following abbreviations are used in this paper:

| | |
|---|---|
| IoT | Internet of Things |
| IIoT | Industrial Internet of Things |
| ICT | Information and Communication Technology |
| API | Application programming interface |
| CapEx | Capital expenditures |
| IaaS | Infrastructure as a service |
| PaaS | Platform-as-a-Service |
| SaaS | Software-as-a-Service |
| WSN | Wireless Sensor Networks |
| RFID | Radio Frequency Identification |
| ML | Machine learning |
| GPS | Global Positioning System |
| SMEs | Small and medium-sized enterprises |
| AI | Artificial Intelligence |

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
