# Peer review of "Cloud-Based IoT Applications and Their Roles in Smart Cities"

_smartcities, doi:10.3390/smartcities4030064_

Round 1

Reviewer 1 Report

Please read the review report and do the necessary correction as need.

Author Response

The author has presented the manuscript “Cloud-Based IoT Applications and their roles in Smart Cities”. I find the work significant and noteworthy for the publication in “Smart Cities” journal. But I observed some minor corrections.

  1. Introduction part is so long, divide it into parts.

Response: Dear Reviewer I have divided the introduction part into sections.

  1. Organization of the paper is missing. Add it at the end of the Introduction section.

Response: Dear Reviewer I have added the organization of the paper at the end of Introduction section.

  1. In Table 1 remove one IoT word in the following reference

Park et al. [19] 2018 The role of IoT (IoT) in smart cities: Technology roadmap-oriented

Approaches

Response: Dear Reviewer I have removed duplicate word.

  1. In line 83, commercial enterprise: should be written as Commercial enterprise:

Response: Dear Reviewer I have corrected it.

  1. In Line 723, correct IoT (IoT). Remove one IoT.

Response: Dear Reviewer I have removed duplicate word.

  1. In line 408, full stop (.) is missing after cloud.

Response: Dear Reviewer I have corrected it.

  1. In line 414, (SaaS) (SaaS). Remove one (SaaS).

Response: Dear Reviewer I have removed duplicate word.

  1. GPS is missing in the Abbreviations.

Response: Dear Reviewer I have added GPS in abbreviations.

  1. References are not written according to the journal guidelines.

Response: References are now revised.

Reviewer 2 Report

The authors dealt with a current issue the paper is well structured, clearly written. - insert representative figures in the introductory part, - - the conclusions must be increased, also referring to future work -the English language is good, check the grammar part, bibliographic suggestions

Autonomous vehicles: An analysis both on their distinctiveness and the potential impact on urban transport systems
Severino, A., Curto, S., Barberi, S., Arena, F., Pau, G. Applied Sciences (Switzerland)this link is disabled,

Author Response

The authors dealt with a current issue the paper is well structured, clearly written. - insert representative figures in the introductory part, - - the conclusions must be increased, also referring to future work -the English language is good, check the grammar part, bibliographic suggestions

Autonomous vehicles: An analysis both on their distinctiveness and the potential impact on urban transport systems Severino, A., Curto, S., Barberi, S., Arena, F., Pau, G. Applied Sciences (Switzerland)

Response: Dear Reviewer, thanks for your comment, I have revised my paper accordingly.

Reviewer 3 Report

This submission explored the cloud based IoT applications and their roles in smart cities.

The survey looks valuable. However,
1.This submission has not sufficiently clarified the novelty of the proposed approach. 
2. This submission misses discussing a few relevant works, such as

  • “Artificial Intelligence Enabled Internet of Things: Network Architecture and Spectrum Access”, IEEE Computational Intelligence Magazine, vol. 15, no. 1 , pp. 44 - 51, Feb. 2020. DOI: 10.1109/MCI.2019.2954643
  • "Energy-Efficient Cooperative Communication and Computation for Wireless Powered Mobile-Edge Computing",  IEEE Systems Journal, Early Access, October 2020, DOI: 10.1109/JSYST.2020.3020474
  • “Internet of Things based Smart Grids Supported by Intelligent Edge Computing”, IEEE Access, vol. 7, pp. 74089-74102, June 2019, DOI: 10.1109/ACCESS.2019.292048

Author Response

This submission explored the cloud based IoT applications and their roles in smart cities.

The survey looks valuable. However,

1.This submission has not sufficiently clarified the novelty of the proposed approach.

Response: Dear Reviewer, thanks for this comment.  In this paper, I have explored the existing cloud-based IoT applications and their roles and importance in smart cities.

  1. This submission misses discussing a few relevant works, such as

“Artificial Intelligence Enabled Internet of Things: Network Architecture and Spectrum Access”, IEEE Computational Intelligence Magazine, vol. 15, no. 1 , pp. 44 - 51, Feb. 2020. DOI: 10.1109/MCI.2019.2954643

"Energy-Efficient Cooperative Communication and Computation for Wireless Powered Mobile-Edge Computing",  IEEE Systems Journal, Early Access, October 2020, DOI: 10.1109/JSYST.2020.3020474

“Internet of Things based Smart Grids Supported by Intelligent Edge Computing”, IEEE Access, vol. 7, pp. 74089-74102, June 2019, DOI: 10.1109/ACCESS.2019.292048

Response: Dear Reviewer, thanks for this comment. In the revised version, I have included the above references.